# Differences in Patient Access to Newly Approved Antibacterial Drugs in EU/EEA Countries

**DOI:** 10.3390/antibiotics13111077

**Published:** 2024-11-12

**Authors:** Anelia Zasheva, Elina Batcheva, Kremena Dimitrova Ivanova, Antoniya Yanakieva

**Affiliations:** 1Department of Infectious Diseases, Military Medical Academy, 1606 Sofia, Bulgaria; 2Department of Pharmacology, Pharmacotherapy and Toxicology, Faculty of Pharmacy, Medical University—Sofia, 1000 Sofia, Bulgaria; elina.bacheva@abv.bg; 3Department of Health Technology Assessment, Faculty of Public Health, Medical University—Sofia, 1527 Sofia, Bulgaria; k.ivanova@foz.mu-sofia.bg (K.D.I.); a.yanakieva@foz.mu-sofia.bg (A.Y.)

**Keywords:** antibiotics, equitable access, antibiotic resistance, one health, antibiotic access

## Abstract

The introduction of antibiotics in the beginning of the 20th century was one of the most important scientific breakthroughs in history. However, in recent decades, the growing threat of antimicrobial resistance (AMR) has shown the limitations of the current research and development programs for new antimicrobial drugs. In the last decade, 20 antibiotics, 7 β-lactam/β-lactamase inhibitor (BL/BLI) combinations and 4 non-traditional antibacterial drugs have been launched worldwide. Methods: This study aimed to assess the time to patient access for new antibacterial drugs in countries in the European Union and the European Economic Area (EU/EEA). Time differences in marketing authorization from the U.S. Food and Drug Agency (FDA) and the European Medicines Agency (EMA) were also described, as well as the availability of each drug in the countries in the EU/EEA according to the national competent authorities. Results: Substantial differences between countries were observed, with no or only one new drug available in some countries. Conclusions: Improving pricing and reimbursement timelines and fostering collaboration between national health authorities and market authorization holders can enhance timely and equitable patient access to new antibacterial treatments in Europe. Equitable and sustainable access to antibacterial drugs is a cornerstone in the battle against AMR.

## 1. Introduction

Antibiotics are chemical compounds that target bacteria and, thus, are intended to treat and prevent bacterial infections. The introduction of antibiotics in clinical use 115 years ago was one of the most important breakthroughs of the 20th century [1]. Their discovery has also made advancement in other fields of medicine possible, such as oncology, surgery and transplantology. The golden era of antibiotic research and discovery (1940–1960) led to a significant reduction in mortality related to infectious diseases [2]. Many of these drugs are still used in clinical practice to this day. However, their effectiveness has decreased because of the growing threat of antibiotic resistance [1].

The discovery and development of antibacterial medicines is closely linked to the discovery of antimicrobial resistance and the different mechanisms that lead to it. In 1940, Abraham and Chain described a strain of *E. coli* producing the enzyme penicillinase which made it resistant to penicillin and its derivates [3]. The first case of methicillin-resistant *S. aureus* (MRSA) was identified in the United Kingdom in 1962 and was described in the United States six years later [4,5]. At the beginning of 1960s, Japanese researchers discovered that resistance can be transferred by genetic elements, later called plasmids, between strains and even between different bacteria, not only vertically, but also horizontally [6,7]. At the same time, multiple reports highlighted the transmission of resistant bacteria from animals to humans due to the overconsumption of antibiotics for growth promotion in cattle breeding [8].

Antimicrobial and particularly antibiotic resistance is one of the most significant challenges facing healthcare systems worldwide nowadays. In countries in the European Union and the European Economic Area, antibiotic-resistant infections resulted in more than 38,000 deaths and more than 1 million years of life lost because of premature death or years lived with disabilities in 2019 alone [9]. Antimicrobial resistance (AMR) also has a significant economic impact, costing the European Union (EU) approximately EUR 1.1 billion annually in additional healthcare costs [10]. According to a review published in 2018, healthcare-associated infections (HAIs) are the third most common adverse event worldwide [11]. Resistant microorganisms are the most frequent cause of HAIs [12]. The importance of AMR and infections caused by multidrug-resistant microorganisms was highlighted by the European Commission in the newly adopted pharmaceutical strategy for Europe because of the decrease in options to treat this type of infections [13].

Antimicrobial resistance is a global issue. However, low- and middle-income countries (LMICs) experience a more serious lack of access to antibiotics and antifungals because many medicines have not yet been registered with national regulatory bodies. This means they cannot be sold or distributed to the patients who need them the most [14,15]. As a result, when the right treatment is not accessible, doctors have to resort to suboptimal treatments, which gives pathogens an opportunity to develop resistance. This is why it is critical to ensure appropriate access to antibacterial drugs. To ensure appropriate access, essential medicines should be available, affordable and accessible, while making sure strong stewardship practices are also implemented. As discussed in other reviews, another issue in these countries is the overuse of the available antibiotics, which has risen even more in recent years and further promotes antimicrobial resistance [16,17,18].

The procedure of the market authorization of innovative medicines in Europe is centralized, and to gain market access, the market authorization holder (MAH) submits the needed documents directly to the European medicine agency (EMA) [19]. After the successful authorization of a medicine by the EMA, the National Health Authorities (NHAs) decide on the pricing and reimbursement. The time to access in different countries differs due to companies’ policy or national priorities. Studying the time to market access and availability within the reimbursement of innovative antimicrobials is important for coordinating the fight against AMR. Access to effective antibacterial drugs is the foundation of a functioning health system and is crucial for responding to emerging health threats and achieving universal healthcare [20].

Hence, our review aims to assess the access to and availability of innovative antibacterials through analyzing the approval dates of innovative antibacterials approved in the last 10 years in the major markets and their availability in countries in the EU/EEA. Time to market access, differences per indication and approval time were collected. This information raises awareness of the variability in patient access across Europe and would help to support EU and national policy makers to reduce inequities.

## 2. Materials and Methods

This is a review focusing on the authorized innovative antibacterials since January 2014 and their market presence. Under innovative antibacterials, we assume new international nonproprietary names (INNs) of antimicrobials authorized as either mono products or fixed-dose combinations (FDCs).

Information about the date of approval was retrieved by analyzing the scientific literature and internet sources, such as company and funding organization websites and the European Medicines Agency (EMA) and U.S. Food and Drug Administration (FDA) webpages. Pharmacological class and therapeutic indications were gathered from the authorization information and the Short Product Characteristics of identified antimicrobial drugs. Access was measured using the time difference in months between the approval date from the FDA and EMA and the market availability of the drugs.

The availability of each antimicrobial in countries in the European Union and European Economic Area (EU/EEA) was identified through the websites of the national competent authorities on pricing and reimbursement, or from the webpages of authorization bodies when reimbursement and pricing information was not available. Detailed information is presented in the Appendix A.

Additionally, we describe the antibacterials included in this review by their classification according to World Health Organization Access, Watch, and Reserve (AWaRe) classification (https://www.who.int/publications/i/item/WHO-MHP-HPS-EML-2023.04 Accessed on 28 August 2024), as well as their presence in the World Health Organization Model List of Essential Drug lists (WHO EML) versions 23 (https://www.who.int/publications/i/item/WHO-MHP-HPS-EML-2023.02 Accessed on 28 August 2024). The AWaRe classification is a tool to monitor antibiotic consumption and the effect of stewardship programs. Antibiotics are classified into three groups, access, watch and reserve, depending on the impact of different antibiotics and antibiotic classes on antimicrobial resistance [21]. The classification is also intended to highlight the appropriate use of antibiotics and is updated every 2 years. The WHO EML lists the most effective and safe medications which meet the most important needs in healthcare systems worldwide [22].

The identified newly approved antibacterial drugs were separated into three groups, mono products, fixed-dose combinations and non-traditional antibacterial drugs, which included monoclonal antibodies and fecal microbiota.

## 3. Results

A total of 31 antimicrobial drugs newly approved worldwide in the last 10 years were identified. Twenty of the identified antibiotics were mono products, seven were fixed-dose combinations and four were non-traditional antibacterial drugs.

### 3.1. Mono Products and Their Approval

In Table 1, all small-molecule antibacterial drugs approved in the last decade are reviewed, and their class, country of first approval, as well as therapeutic indications and ATC code are provided where available.

A total of nine drugs received approval from the EMA. Delamanid and Eravacycline are the only antibacterial drugs which received approval of use from the EMA first and then were approved by the FDA. A total of 12 antibiotics were first approved in the USA. Six of the antibiotics were approved outside of Europe and the USA, and none of them have approval in countries in the EU/EEA or the States. Some of the MAHs outside of Europe and the USA have declared their intention to pursue global approval for their antimicrobial products. To accelerate the global approval of contezolid, the MAH MicuRx intends on conducting an international phase III study, as well as a phase III study in the USA to assess the efficacy and safety of intravenous contezolid acefosamil (MRX-4; a prodrug of contezolid) followed by oral contezolid in ABSSSIs [24,25].

None of the authorized 20 products are first-in-class. They belong to the well-known pharmacological classes of fluoroquinolones (*n* = 5), tetracyclines (*n* = 3), nitroimidazoles (*n* = 3), glycopeptides (*n* = 2), oxazolidinone (*n* = 2), cephalosporines (*n* = 2) and one INN from the groups of quinolones, aminoglycosides, and pleuromutilines. The majority are antibacterial drugs for the treatment of acute or chronic SSSIs (*n* = 9), while two are new drugs for the treatment of tuberculosis, and the rest are intended for the treatment of urinary tract and gynecological infections, sinusitis, otitis, pneumonia and other conditions.

### 3.2. Fixed-Dose Combination Drugs and Their Approval

In Table 2, all fixed-dose combination antibacterial drugs approved in the last decade are reviewed, and their class, country of first approval, as well as therapeutic indications and ATC code are provided where available.

A total of six drugs were authorized by the EMA. Among the seven identified fixed-dose combinations, one is authorized only by the FDA (durlobactam + sulbactam) and one only by the EMA (aztreonam + avibactam)—Table 2. Emblaveo, the combination of aztreonam and avibactam, is the beta-lactam and beta-lactamase inhibitor (BL/BLI) combination which does not have approval by the FDA.

Aztreonam is not a newly discovered drug. It was first approved in the USA and Europe in 1986 and was first used for intrabdominal, respiratory and urinary tract infections [26]. In the last three decades, its use was limited because of the increased number of infections caused by bacteria producing extended β-lactamase. The use of aztreonam in combination with a β-lactamase inhibitor shows good in vitro efficacy against metallo-β-lactamase-producing Gram-negative bacteria [27]. This is the second BL/BLI combination which is active against those isolates.

Exblifep, the combination of enmetazobactam and cefepime, is the most recently approved drug from both the FDA and EMA and is also active against metallo-β-lactamase-producing bacteria [28].

The only combination drug which is not approved by the EMA is Xacduro. This drug combines durlobactam and sulbactam and is the first and only (for now) medication indicated for hospital-acquired pneumonia caused by *Acinetobacter* complex [29]. Infections caused by multi-drug resistant (MDR) *Acinetobacter* species are among the most serious threats to human health and lead to significant mortality and morbidity, which is why the authorization of this FDC is of great importance [30,31]. These bacteria are the cause of a variety of infections, including but not limited to surgical site infections, ventilator-associated pneumonia, complicated urinary tract infections and others, in both immunocompetent and immunocompromised individuals [32]. This organism is resistant to the majority of the antibiotic classes, including cephalosporins, fluoroquinolones, aminoglycosides and tetracyclines [33,34].

According to their pharmacological group, beta-lactam and beta-lactamase inhibitors alone or with diazabicyclooctane (DBO) prevail (*n* = 6 out of 7). These combinations have showed better pharmacokinetics/pharmacodynamics profiles and greater efficacy, as well as other advantages. Regarding the therapeutic indications, five of the FDCs are against Gram-negative G (-) urinary or abdominal infectious, and almost all are indicated for the treatment of hospital-acquired and ventilator-associated pneumonia (HAP/VAP).

### 3.3. Other Antibacterial Drugs and Their Approval

Table 3 gives an overview of non-traditional antibacterial drugs approved in the last decade, their class, country of first approval, as well as therapeutic indications and ATC code where available.

Obiltoxaximab is approved by both leading agencies and is a monoclonal antibody used for pre- and postexposure to inhalation anthrax [35]. Obiltoxaximab, a human immunoglobulin G1 (IgG1) mAb, exerts its effect by binding to and neutralizing protective antigen, preventing it from binding to cellular receptors [35,36]. Bacillus anthracis is one of the potential bioterrorist agents associated with high morbidity and mortality [37].

Bezlotoxumab is a human monoclonal antibody which binds directly to C. difficile toxin B and prevents intestinal epithelial damage and colitis [38,39]. Furthermore, this monoclonal antibody decreases the recurrence of CDI because it attributes to the early reconstitution of gut microbiota [40].

The other two therapies, Rebyota and Vowst, are not approved by the EMA. They are both single-dose, rectally administered, microbiota-based live biotherapeutics approved for use to prevent the recurrence of CDI in individuals 18 years and older following antibiotic treatment for recurrent CDI [41,42].

### 3.4. Newly Approved Antibacterials and WHO AWaRe Classification and WHO EML

As listed in Table 4, 13 of the mono products and four of the FDCs are listed in the WHO AWaRe Classification. The majority (*n* = 12) are classified as reserve, and the rest (*n* = 5) as watch. According to the classification, antibiotics in the watch group are recommended only when no other options are possible or available and should be used carefully [43]. The medications listed in the reserve group are last-line options and are used for multidrug-resistant infections. Six of the newly approved antibiotics are present in the version from 2023 of the WHO List of Essential Medicines.

### 3.5. Time Difference in Approval

The time difference between authorization from the FDA and EMA was calculated in months for the antibacterial medicines approved by both agencies (Table 5).

The median time difference between the approval by both agencies in months was 12.5 months (IQR 1-55). Eravacycline and Exblifep are the antibacterial drugs with shortest difference in time of approval: they were authorized by the EMA only one month after their authorization by the FDA. Obiltoxaximab is the product which was approved by the EMA 55 months after its first approval from the FDA.

It should be noted that time difference between the submission of documents by the market authorization holder and the issue of marketing authorization was not taken into account.

### 3.6. Availability of Newly Approved Antibacterials in the Countries of EU/EEA

Regarding the time between EMA approval and patients’ antibiotics access in EU/EEA countries, we found considerable variations between medicines and countries. Detailed information about each antibacterial drug and each country in the EU/EEA is available in Appendix A. All of the drugs in the present review received authorization by a centralized procedure.

None of the antibacterial medicines presented above are available in Iceland, while only one is available in Malta and Latvia. In Malta, only dalbavancin is available, and in Latvia, the BL/BLI combination drug avibactam + ceftazidime is marketed. In Lithuania, the commercialized medicines are avibactam+ceftazidime and vaborbactam + meropenem.

Only four medicines are commercialized in Luxembourg and Cyprus. In both countries, ceftolozane + tazobactam, aztreonam + avibactam and avibactam + ceftazidime are available. In Cyprus, the other available drug is relebactam + imipenem+cilastatin (Recarbrio), while in Luxembourg, this is vaborbactam + meropenem.

On the other hand, all antibacterial medicines are commercialized in Austria, Ireland, Italy and Slovakia. In Estonia, Germany, the Netherlands, Poland and Liechtenstein, only one drug is not marketed (Table 6).

Countries in which all innovative antibiotics are available differ in terms of their economic development. Austria, Ireland and Italy are economically well developed, while Slovakia falls within the group of medium-income countries. Further analysis should be carried out to explain the differences in availability.

As shown in Table 7, the FDCs are available in the highest number of countries in the EU/EEA. For example, avibactam+ceftazidime is available in *n* = 27, followed by ceftolozane+tazobactam (*n* = 25). The monoclonal antibody obiltoxaximab was available in only nine countries (according to the data gathered in August 2024 for this article), before its marketing authorization was withdrawn at the request of the MAH in September 2024.

## 4. Discussion

In this review, we report on patient access to innovative medicines in countries in the EU/EEA. To our knowledge, this is the first study focusing on patient access to new antibacterial treatments in the EU/EEA.

Our review found considerable differences between medicines and countries despite the centralized procedure of market authorization of the EMA in the EU. There is also a delay in the authorization of the antibacterial drugs in Europe compared to the USA. Based on our study, we can suppose that MAHs prefer to authorize their medicines in the USA first.

The observed heterogeneity of access across European countries is surprising, although differences in access to medicines based on countries’ healthcare systems are to be expected. The time to access in our study is likely impacted by national trends in the pricing and reimbursement system. Delays in patients’ access may also be due to the strategies of the MAH for the pricing and launch of their medicines, as well as the reimbursement processes of the national health authorities (NHAs).

The use of pricing and launch strategies which maximize the financial gain for the MAH may be one of the reasons for delays and differences in patient access in different European countries [44,45]. In addition, bigger European pharmaceutical markets like Italy, France and the Netherlands are preferred for the marketing of innovative medications by MAHs [46]. These strategies may be a reason for delayed patient access in countries with a smaller market size, because of the smaller target population which would result in less profits [47].

In a systemic analysis on the global burden of bacterial antimicrobial resistance in the period 1990–2021, the GBD 2021 Antimicrobial Resistance Collaborators forecasts that in 2050, there will be 1.91 million annual deaths attributable to AMR worldwide, while the number of deaths associated with AMR would be 8.22 million [48]. According to the same analysis and forecast, by 2050, the death rates per 100,000 attributable to AMR in Europe will rise in the countries of the Balkan Peninsula, Poland, Italy, Spain, Portugal and Croatia. It will remain significantly low in Scandinavian countries [48]. According to the report on AMR surveillance in Europe published in 2023 by the European Centre for Disease Prevention and Control (ECDC) and WHO/Europe, which features data from 2021, countries in Southern and Eastern Europe reported higher levels of antimicrobial resistance compared to Northern and Western Europe [49]. As our study shows, there is a heterogeneity in patients’ access to newly authorized antimicrobials in the countries in these two European regions. For example, in Bulgaria, the Czech Republic and Slovenia, the number of marketed drugs is *n* = 5, while in Italy and Slovakia, all of them are marketed. The data in the report also show a significant increase in isolates of carbapenem-resistant *Acinetobacter* species, as well as third-generation cephalosporin-resistant *K. pneumoniae* [49]. Additionally, in May 2024, the ECDC published their point prevalence survey of healthcare-associated infections and antimicrobial use in European acute care hospitals—2022–2023. The estimated number of patients in acute care hospitals which acquire at least one healthcare-associated infection (HAI) caused by multidrug-resistant bacteria annually is 262,833 people [50].

The crisis in the research and development of new antibacterial drugs is well known and widely discussed [51,52]. Here, we describe serious limitations on commercial launches and patients’ access in high- and middle-income countries [53]. While antimicrobial stewardship is important in tackling the AMR pandemic, equitable access to novel drugs when needed also has a role to play in diminishing the burden of deaths and disabilities caused by infections with multidrug-resistant bacteria.

Our research, however, has a few limitations. One major setback is the authenticity of the presented information as it is gathered by publicly accessible resources from the National Medicines Registers. This implies that the collected data are subject to change. The present study does not include information on use of the antimicrobials presented above as part of clinical trials. Another limitation is that it does not use data from the MAHs on the time of market access in each country in the EU/EEA after receiving authorization from EMA. Further research is needed to determine the causes of the heterogeneity of patients’ access to antibacterials presented in this study. To better understand the reasons behind it, information on the pricing and reimbursement processes of NAHs can be explored.

## Figures and Tables

**Table 1 antibiotics-13-01077-t001:** Small-molecule antibacterial drugs approved worldwide in the last decade.

	Drug Name	Class	Country of First Approval	Date of FDA Approval	Date of EMA Approval	Therapeutic Indication
1.	delamanid (J04AK06)	nitroimidazole	Europe	not approved	28 April 2014	TB
2.	dalbavancin (J01XA04)	glycopeptide	USA	23 May 2014	19 February 2015	G+ve SSSI
3.	oritavancin (J01XA05)	glycopeptide	USA	6 August 2014	19 March 2015	G+ve SSSI
4.	tedizolid phosphate (prodrug) (J01XX11)	oxazolidinone	USA	20 June 2014	23 March 2015	G+ve SSSI
5.	nemonoxacin (J01MB08)	quinolone	Taiwan	not approved	not approved	G+ve/G−ve cSSSI
6.	morinidazole	nitroimidazole	China	not approved	not approved	G+ve/G−ve gynecological and suppurative appendicitis
7.	finafloxacin	fluoroquinolone	USA	17 December 2014	not approved	acute otitis externa
8.	zabofloxacin	fluoroquinolone	Republic of Korea	not approved	not approved	G+ve/G−ve CABP
9.	delafloxacin (J01MA23)	fluoroquinolone	USA	19 June 2017	16 December 2019	G+ve/G−ve ABSSSI and CABP
10.	plazomicin (J01GB14)	aminoglycoside	USA	25 June 2018	not approved	G−ve UTI
11.	eravacycline (J01AA13, J01AA20)	tetracycline	Europe	28 August 2018	26 July 2018	G+ve/G−ve IAI
12.	omadacycline (J01AA15, J01AA20)	tetracycline	USA	2 October 2018	not approved	G+ve/G−ve CABP and ABSSSI
13.	sarecycline (J01AA14, J01AA20)	tetracycline	USA	2 October 2018	not approved	G+ve acne
14.	pretomanid (J04AK08)	nitroimidazole	USA	14 August 2019	31 July 2020	TB
15.	lefamulin (J01XX12)	pleuromutilin	USA	19 August 2019	27 July 2020	G+ve/G−ve CABP
16.	lascufloxacin (J01MA25)	fluoroquinolone	Japan	not approved	not approved	G+ve/G−ve CABP and sinusitis
17.	cefiderocol (J01DI04)	cephalosporin siderophore	USA	14 November 2019	23 April 2020	G−ve cUTI and bacterial infections
18.	levonadifloxacin; alalevonadifloxacin (prodrug) (J01MA24)	fluoroquinolone	India	not approved	not approved	G+ve/G−ve ABSSSI
19.	contezolid	oxazolidinone	China	not approved	not approved	G+ve cSSSI
20.	ceftobiprole medocaril	cephalosporin	USA	3 April 2024	not approved	SAB, G+ve/G−ve ABSSSI and CABP

ABSSSI, acute bacterial skin and skin structure infection; CABP, community-acquired bacterial infection; SSSI, skin and skin structure infection; cSSSI, complicated SSSI; UTI, urinary tract infection; cUTI, complicated UTI; TB, tuberculosis; USA United States of America. This table is adapted from the article “Antibiotics in the clinical pipeline as of December 2022” by Butler, M.S., Henderson, I.R., Capon, R.J. et al. [23] The article is licensed under the Creative Commons Attribution 4.0 International License. © 2023 Butler, M.S., Henderson, I.R., Capon, R.J. et al.

**Table 2 antibiotics-13-01077-t002:** Fixed-dose combination drugs approved worldwide in the last decade.

	Drug Name	Class	Country of First Approval	Date of FDA Approval	Date of EMA Approval	Therapeutic Indication
1.	Zerbaxa: ceftolozane + tazobactam (J01DI54)	BL + BLI	USA	19 December 2024	18 September 2015	G−ve cUTI, cIAI and HAP/VAP
2.	Avycaz: avibactam + ceftazidime (J01DD52)	DBO BLI + BL	USA	25 May 2015	23 June 2016	G−ve cUTI, cIAI and HAP/VAP
3.	Vabomere/Vaborem: vaborbactam + meropenem (J01DH52)	boronate BLI + BL	USA	29 August 2017	20 November 2018	G−ve cUTI, cIAI and HAP/VAP
4.	Recarbrio: relebactam + imipenem + cilastatin (J01DH56)	DBO BLI + BL+ renal dehydropeptidase inhibitor	USA	17 July 2019	13 February 2020	G−ve cUTI, cIAI and HAP/VAP
5.	Xacduro: durlobactam + sulbactam	DBO BLI + clavulanic acid	USA	23 May 2023	not approved	MDR *Acinetobacter* infections
6.	Exblifep: enmetazobactam + cefepime (J01DE51)	clavulanic acid + cephalosporin	USA	22 February 2024	21 March 2024	cUTI, HAP/VAP
7.	Emblaveo: aztreonam + avibactam (J01DF51)	BL + BLI	Europe	not approved	23 April 2024	G−ve cUTI, cIAI and HAP/VAP

BLI, β-lactamase inhibitor; BL, β-lactam; DBO, diazabicyclooctane; HAP/VAP, hospital/ventilator-associated pneumonia; cIAI, complicated intra-abdominal infection; cUTI, complicated UTI; USA, United States of America. This table is adapted from the article “Antibiotics in the clinical pipeline as of December 2022” by Butler, M.S., Henderson, I.R., Capon, R.J. et al. [23] The article is licensed under the Creative Commons Attribution 4.0 International License. © 2023 Butler, M.S., Henderson, I.R., Capon, R.J. et al.

**Table 3 antibiotics-13-01077-t003:** Non-traditional antibacterial drugs approved worldwide in the last decade.

	Drug Name	Class	Country of First Approval	Date of FDA Approval	Date of EMA Approval	Therapeutic Indication
1.	obiltoxaximab (J06BC04)	mAb	USA	18 March 2016	18 November 2020	G+ve anthrax
2.	bezlotoxumab (J06BC03)	mAb	USA	21 October 2016	18 January 2017	G+ve anthrax
3.	Rebyota (RBX2660)	microbiome	USA	30 November 2022	not approved	G+ve CDI
4.	Vowst (SER-109)	microbiome	USA	26 April 2023	not approved	G+ve CDI

CDI, C. difficile infection; G+ve, Gram-positive bacteria; mAb, monoclonal antibody; USA, United States of America. This table is adapted from the article “Antibiotics in the clinical pipeline as of December 2022” by Butler, M.S., Henderson, I.R., Capon, R.J. et al. [23] The article is licensed under the Creative Commons Attribution 4.0 International License. © 2023 Butler, M.S., Henderson, I.R., Capon, R.J. et al.

**Table 4 antibiotics-13-01077-t004:** Newly approved antibacterial drugs classified according to the WHO AWaRe Classification and their presence in the WHO EML.

	Drug Name	Class	WHO AWaRe Group	Listed on EML/EMLc 2023
1.	delamanid (J04AK06)	nitroimidazole	Not listed	No
2.	dalbavancin (J01XA04)	glycopeptide	Reserve	No
3.	oritavancin (J01XA05)	glycopeptide	Reserve	No
4.	tedizolid phosphate (prodrug) (J01XX11)	oxazolidinone	Reserve	Yes
5.	nemonoxacin (J01MB08)	quinolone	Watch	No
6.	morinidazole	nitroimidazole	Not listed	No
7.	finafloxacin	fluoroquinolone	Not listed	No
8.	zabofloxacin	fluoroquinolone	Not listed	No
9.	delafloxacin (J01MA23)	fluoroquinolone	Watch	No
10.	plazomicin (J01GB14)	aminoglycoside	Reserve	Yes
11.	eravacycline (J01AA13, J01AA20)	tetracycline	Reserve	No
12.	omadacycline (J01AA15, J01AA20)	tetracycline	Reserve	No
13.	sarecycline (J01AA14, J01AA20)	tetracycline	Watch	No
14.	pretomanid (J04AK08)	nitroimidazole	Not listed	No
15.	lefamulin (J01XX12)	pleuromutilin	Reserve	No
16.	lascufloxacin (J01MA25)	fluoroquinolone	Watch	No
17.	cefiderocol (J01DI04)	cephalosporin siderophore	Reserve	Yes
18.	levonadifloxacin; alalevonadifloxacin (prodrug) (J01MA24)	fluoroquinolone	Watch	No
19.	contezolid	oxazolidinone	Not listed	No
20.	ceftobiprole medocaril	cephalosporin	Not listed	No
21.	Zerbaxa: ceftolozane + tazobactam (J01DI54)	BL + BLI	Reserve	Yes
22.	Avycaz: avibactam + ceftazidime (J01DD52)	DBO BLI + BL	Reserve	Yes
23.	Vabomere/Vaborem: vaborbactam + meropenem (J01DH52)	boronate BLI + BL	Reserve	Yes
24.	Recarbrio: relebactam + imipenem + cilastatin (J01DH56)	DBO BLI + BL + renal dehydropeptidase inhibitor	Reserve	No
25.	Xacduro: durlobactam + sulbactam	DBO BLI + clavulanic acid	Not listed	No
26.	Exblifep: enmetazobactam + cefepime (J01DE51)	clavulanic acid + cephalosporin	Not listed	No
27.	Emblaveo: aztreonam + avibactam (J01DF51)	BL + BLI	Not listed	No

**Table 5 antibiotics-13-01077-t005:** Time difference in approval by FDA and EMA in months.

	Drug Name	Time Difference in Months
1.	dalbavancin	8
2.	oritavancin	7
3.	tedizolid phosphate (prodrug)	10
4.	delafloxacin	30
5.	eravacycline	1
6.	pretomanid	12
7.	lefamulin	12
8.	cefiderocol	6
9.	Zerbaxa: ceftolozane + tazobactam	9
10.	Avycaz: avibactam + ceftazidime	13
11.	Vabomere/Vaborem: vaborbactam + meropenem	14
12.	Recarbrio: relebactam + imipenem + cilastatin	7
13.	Exblifep: enmetazobactam + cefepime	1
14.	obiltoxaximab	55
15.	bezlotoxumab	3

**Table 6 antibiotics-13-01077-t006:** Number of marketed drugs in each country in EU/EEA.

	Country	Number of Marketed Drugs
1	Austria	15
2	Belgium	4
3	Bulgaria	5
4	Croatia	10
5	Cyprus	4
6	Czech Republic	5
7	Denmark	6
8	Estonia	14
9	Finland	11
10	France	11
11	Germany	14
12	Greece	13
13	Hungary	13
14	Ireland	15
15	Italy	15
16	Latvia	1
17	Lithuania	2
18	Luxembourg	4
19	Malta	1
20	Netherlands	14
21	Poland	14
22	Portugal	13
23	Romania	6
24	Slovakia	15
25	Slovenia	5
26	Spain	10
27	Sweden	13
28	Iceland	0
29	Liechtenstein	14
30	Norway	6

**Table 7 antibiotics-13-01077-t007:** Number of countries where each antibacterial drug is marketed.

	Drug Name	Number of Countries Where It is Marketed
1.	dalbavancin	20
2.	oritavancin	17
3.	tedizolid phosphate (prodrug)	17
4.	delafloxacin	16
5.	eravacycline	16
6.	pretomanid	14
7.	lefamulin	13
8.	cefiderocol	19
9.	Zerbaxa: ceftolozane + tazobactam	25
10.	Avycaz: avibactam + ceftazidime	27
11.	Vabomere/Vaborem: vaborbactam + meropenem	21
12.	Recarbrio: relebactam + imipenem + cilastatin	23
13.	Exblifep: enmetazobactam + cefepime	12
14.	Emblaveo: aztreonam + avibactam	16
15.	obiltoxaximab	Marketing authorization withdrawn in September 2024
16.	bezlotoxumab	17

## Data Availability

The authors confirm that the data supporting the findings of this study are available within the article and its Appendix A.

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
