# Peer review of "Differences in Patient Access to Newly Approved Antibacterial Drugs in EU/EEA Countries"

_antibiotics, 2024, doi:10.3390/antibiotics13111077_

Round 1

Reviewer 1 Report

Comments and Suggestions for Authors

I read with interest the paper titled "Differences in time to patient access to newly approved anti-bacterial drugs in the countries of EU/EEA"

The paper reveals a study of the important diferences in market access. 

I have only minor comments to add:

1. XXth century could be 20th Century. 

2. Any founds that could clarify the importance (of coverage) of population using those drugs? Are we expecting a lot of people using?

3. Why some of them are not approved internationally with consensus?

4. Why Figure 1 is only focused on Europe? And if this is a main objective, please add the type of introduction procedure connected with the drug (centralized, descentralized, national procedures?)

5. Fig 2 type of graph, makes no sense. This is a line plot to show trends. In such case, authors wish to compare different drugs, so a different plot should be applied. 

6. Overall discussion is poor and should be further improved. Why are the medicines approved firstly in some countries compared to another? Whats the reasons? strict regulation? Limitations and strengths should be added. 

Author Response

Dear Reviewer,

Thank you for your comments!

Comment 1: XXth century could be 20th Century. 

Response 1: We have corrected it to 20th century.

Comment 2: Any founds that could clarify the importance (of coverage) of population using those drugs? Are we expecting a lot of people using?

Response 2: Findings on the current AMR situation in Europe and forecasts for the situation in 25 years time have been added in the Discussion.

Comment 3: Why some of them are not approved internationally with consensus?

Response 3: We have added information about MAHs who have plans for international authorization. However, this information is not available for all and it is possible that not all of them have plans on submitting documents for market authorization outside of the first country of approval.

Comment 4: Why Figure 1 is only focused on Europe? And if this is a main objective, please add the type of introduction procedure connected with the drug (centralized, descentralized, national procedures?)

Response 4: It is only focused on Europe because it is in the subsection titled 'Availability of newly approved antibacterials in the countries of EU/EEA'. Type of authorization procedure is also added.

Comment 5: Fig 2 type of graph, makes no sense. This is a line plot to show trends. In such case, authors wish to compare different drugs, so a different plot should be applied. 

Response 5: We have converted figures 1 and 2 into tables.

Comment 6: Overall discussion is poor and should be further improved. Why are the medicines approved firstly in some countries compared to another? Whats the reasons? strict regulation? Limitations and strengths should be added. 

Response 6: We have improved the discussion part and we have described the limitations of the study.

Reviewer 2 Report

Comments and Suggestions for Authors

There is great problem due to bacterial resistances worldwide, therefore this manuscript can make a contribution in understanding and trying to solve some aspects in the regulatory and process of obtention of new antibiotics

There are several things that should be commented and improved in the manuscript.

-          Title: I consider that the term “time” should no appear in the title. In the manuscript the only data available are the difference in time approval between FDA and EMA. Even though the authors include the problem of time between EMA approval and the commercialization in the different countries, they do not show results.

-          The method section should be section 2, before the results. Links to WHO Access Watch and Reserve, and Essential Medicines should be referenced at this point.

-          Results:

o   In Table 1, antibiotic class should include the ATC

o   Tables should be inserted at the end of the paragraph

o   Paragraph of lines 190 to 196, it is not clear, it should be rewritten

o   The information of Figures 1 and 2 would be better in table format, especially Figure2.

o   In these Figures are not included all the countries of the European Union, even thought in the supplementary material there is this information.

Author Response

Dear Reviewer,

Thank you for your comments!

Comment 1: Title: I consider that the term “time” should no appear in the title. In the manuscript the only data available are the difference in time approval between FDA and EMA. Even though the authors include the problem of time between EMA approval and the commercialization in the different countries, they do not show results.

Response 1: We have removed 'time' from the title.

Comment 2: The method section should be section 2, before the results. Links to WHO Access Watch and Reserve, and Essential Medicines should be referenced at this point.

Response 2: We have moved the method section after the introduction and have added the references to WHO AWaRe and WHO EML in this part of the article.

Comment 3: 

  • Results:
    • In Table 1, antibiotic class should include the ATC
    • Tables should be inserted at the end of the paragraph
    • Paragraph of lines 190 to 196, it is not clear, it should be rewritten
    • The information of Figures 1 and 2 would be better in table format, especially Figure2.
    • In these Figures are not included all the countries of the European Union, even thought in the supplementary material there is this information.

Response 3: ATC code has been added to table 1 and all tables have been moved to the end of the paragraph. The paragraph of lines 190-196 has been rewritten. Figures 1 and 2 have been converted in tables and now include information on all countries of EU/EEA.

Round 2

Reviewer 2 Report

Comments and Suggestions for Authors

The authors have made improvements in the manuscript following the suggestions of the review.